# Hardware Development and Evaluation of Multihop Cluster-Based Agricultural IoT Based on Bluetooth Low-Energy and LoRa Communication Technologies

**DOI:** 10.3390/s24186113

**Published:** 2024-09-21

**Authors:** Emmanuel Effah, George Ghartey, Joshua Kweku Aidoo, Ousmane Thiare

**Affiliations:** 1Computer Science and Engineering Department, University of Mines and Technology, Tarkwa P.O. Box 237, Ghana; georgeghartey2@gmail.com (G.G.); ce-jaidoo@st.umat.edu.gh (J.K.A.); 2Department of Informatics, Gaston Berger University, Saint-Louis PB 234, Senegal; ousmane.thiare@ugb.edu.sn

**Keywords:** Internet of Things (IoT), agricultural IoT (Agri-IoT), multihop cluster-based agricultural IoT (MCA-IoT)

## Abstract

In this paper, we present the development and evaluation of a contextually relevant, cost-effective, multihop cluster-based agricultural Internet of Things (MCA-IoT) network. This network utilizes commercial off-the-shelf (COTS) Bluetooth Low-Energy (BLE) and LoRa communication technologies, along with the Raspberry Pi 3 Model B+ (RPi 3 B+), to address the challenges of climate change-induced global food insecurity in smart farming applications. Employing the lean engineering design approach, we initially implemented a centralized cluster-based agricultural IoT (CA-IoT) hardware testbed incorporating BLE, RPi 3 B+, STEMMA soil moisture sensors, UM25 m, and LoPy low-power Wi-Fi modules. This system was subsequently adapted and refined to assess the performance of the MCA-IoT network. This study offers a comprehensive reference on the novel, location-independent MCA-IoT technology, including detailed design and deployment insights for the agricultural IoT (Agri-IoT) community. The proposed solution demonstrated favorable performance in indoor and outdoor environments, particularly in water-stressed regions of Northern Ghana. Performance evaluations revealed that the MCA-IoT technology is easy to deploy and manage by users with limited expertise, is location-independent, robust, energy-efficient for battery operation, and scalable in terms of task and size, thereby providing a versatile range of measurements for future applications. Our results further demonstrated that the most effective approach to utilizing existing IoT-based communication technologies within a typical farming context in sub-Saharan Africa is to integrate them.

## 1. Introduction

In addition to being time-consuming, labor-intensive, inefficient, and unreliable, rainfall-dependent agricultural practices are unable to address the current challenges of food insecurity and the associated threats of unemployment arising from the adverse effects of climate change and the growing global population on agriculture, the primary global employer [1,2,3,4,5,6]. For example, in Africa, where this research is focused, traditional agricultural production seasons have been disrupted by droughts induced by climate change. According to the International Monetary Fund (IMF) statistics on the impact of climate change on food insecurity in sub-Saharan Africa (SSA) in 2023, droughts attributable to climate change accounted for over 20 percent of the region’s food insecurity [7], a figure that could double if smart farming technologies do not receive adequate research attention. Since 2022, the number of individuals affected by regional food insecurity has increased from 101 million in 2019 to 145 million [7].

Fortunately, wireless sensor network-based agricultural Internet of Things (WSN-based Agri-IoT) technology has emerged as a promising solution for enhancing resource optimization, remote monitoring, and farm automation. This technology utilizes sampled data on micro-climatic parameters, physical conditions, livestock locations and conditions, and farm activities through diverse wirelessly connected electronic devices (known as sensor nodes (SNs)), systems, and platforms. Agri-IoT technology not only facilitates the management of farming processes, resources, and remote control (i.e., farm automation) but also improves crop quality and production capacity by ensuring that resources such as water, fertilizers, and pesticides are applied at optimal times and under appropriate environmental conditions [4,5,8].

This situation necessitates a paradigm shift in agricultural practices in Africa. The most promising solution must be a robust, affordable, autonomous, and contextually relevant Agri-IoT technology that meets the critical design expectations illustrated in Figure 1.

Currently, the benchmarking Agri-IoT testbed solutions referenced in [3,5,11,12,13,14,15,16], as depicted in Figure 2b, rely on fixed support systems (such as wired sensor networks, servers, and gateway backbones), intricate event-routing architectures/protocols, and costly communication technologies (e.g., Wi-Fi/Wi-Fi–cellular communication technologies) for sensor network communication. These factors render them expensive, location-restricted, energy-inefficient, and overly complex for non-experts and smallholder farmers, who represent over two-thirds of the economically active global population [2,6]. Furthermore, many of these solutions have been validated under indoor conditions [17,18,19,20,21,22,23,24,25,26,27,28,29,30], making their performance indicators less applicable for real-world assessment [3,31]. Although some testbeds operate in outdoor settings [3,5,14,15], their setup and management via centralized or flooding-based routing protocols with wired backbones are often time-consuming, location-constrained, and capital-intensive, especially as the network and SN count increase. Moreover, these solutions typically require on-farm electricity and internet access, which are generally unavailable in many African regions. Consequently, a significant gap remains between the theoretical design of these technologies and their practical performance in real-world settings. Although cluster-based Agri-IoT implementations using multihop architectures [32,33,34,35,36,37] and emerging low-power, wireless communication technologies such as Bluetooth Low-Energy (BLE) and LoRa have demonstrated significant potential to address the technical challenges associated with WSN-based Agri-IoT [4,8,38], they have not been adequately explored in real-world testbed implementations and hardware evaluations due to several technical challenges:

There is a lack of robust, low-power, flexible, location-independent, low-cost, and stable real-world testbed architectural frameworks for multihop cluster-based Agri-IoT (MCA-IoT) that utilize freely available, low-power wireless communication standards such as BLE and LoRa. Such frameworks are needed to facilitate the easy deployment and wireless management of networks without relying on expensive fixed infrastructure. Despite substantial research in this domain, many solutions remain prohibitively costly and impractical for small-scale farms [20].There is an urgent need for comprehensive reference documents that provide detailed real-world accounts of experiences from the design of custom-built multihop-based SNs to their deployment and performance assessment under indoor and outdoor conditions. Investigations are needed to evaluate how the benefits of cluster-based multihop architectures materialize in custom-built MCA-IoT networks.The impact of outdoor environmental conditions (such as humidity, temperature, dust concentration, rainstorms, and crop obstructions) on radio communication (including the effective range and link quality of BLE and LoRa) in typical sub-Saharan African (SSA) settings requires further study.

This paper presents a detailed examination of a real-world application and evaluation of an MCA-IoT network, utilizing freely available wireless communication technologies such as LoRa and BLE. This evaluation is conducted through a realistic multihop architecture, as discussed in [4,8,10,17], and is validated under indoor and outdoor environmental conditions. Specifically, this study introduces a WSN-specific MCA-IoT testbed that encompasses the following elements:A custom-designed, low-power, robust, contextually relevant, and task-scalable SN and MCA-IoT network based on LoRa and BLE 4.2 wireless communication technologies. This configuration ensures that the network is location-independent, user-friendly, easily deployable, and capable of being wirelessly managed by non-experts in IoT without costly fixed support systems.A comprehensive account of the MCA-IoT network framework, detailing the successes and challenges encountered during the design, deployment, and operation of SNs, as well as cloud data storage. This documentation aims to serve as a valuable reference for the Agri-IoT community.An empirical assessment of the practical realization of the theoretical benefits associated with the cluster-based architecture.An observation of the effects of climatic extremes, physical obstructions, and variations in SN power on radio connectivity and packet loss.

The structure of this paper is as follows. Section 2 and Section 3 provide a systematic review of the relevant literature and a theoretical overview of the proposed approach, respectively. Section 4 details the design of the testbed. Section 5 discusses the validation experiments and their results, and Section 6 presents a discussion of the findings and suggestions for future work.

## 2. Background and Synthesis of Related Literature

### 2.1. Background

Over the past decade, human survival has increasingly been jeopardized by factors such as population growth, climate change, and dwindling water resources, leading to significant food insecurity [37]. In response to these challenges, the development of smart and eco-friendly Agri-IoT technology has emerged. This technology is designed to enhance precision farming and greenhouse management, thereby improving food production and crop quality on limited land. By leveraging wireless sensor networks (WSNs), Agri-IoT represents a shift from personal computing to ubiquitous computing, offering enhanced flexibility and simplicity for remote monitoring, system control, and environmental management [3,39,40].

The WSN-based Agri-IoT system, as illustrated in Figure 3, operates as a feedback control mechanism for comprehensive farm management. It facilitates real-time monitoring, data sampling, resource optimization, and automation of agricultural operations, such as precision irrigation and disease management [4]. This system relies on a network of battery-powered, wirelessly connected SNs equipped with sensing, processing, and communication capabilities [39,41,42]. These SNs are spatially distributed and self-configured to enable efficient remote sensing, surveillance, and control through automated data collection and processing [43,44].

Figure 2 shows that a typical WSN-based Agri-IoT network includes field-deployed SNs, a base station (BS) or gateway, and IoT cloud servers that host data analytics engines and user control applications. This architecture allows users to monitor farm conditions and execute control actions via software interfaces such as web pages or custom apps. The routing architectures employed in WSN-based Agri-IoT include a cluster-based routing architecture, a centralized routing architecture, and a flooding-based routing architecture, as illustrated in Figure 4. The system’s ability to provide real-time monitoring and control is a significant advancement over traditional farming practices.

Despite these advancements, Agri-IoT technology has not fully realized its potential due to several challenges. Existing solutions often suffer from high costs, location dependence, and complexity in deployment and management. Additionally, they rely on centralized or mesh-like communication architectures that are neither scalable nor fault-tolerant, as illustrated in Figure 5. Agri-IoT systems must address unique constraints such as battery power limitations, cost efficiency, and the need for fault tolerance, making the choice of routing architecture critical for overcoming these challenges [4,8,45,46,47,48,49].

**Figure 4 sensors-24-06113-f004:**
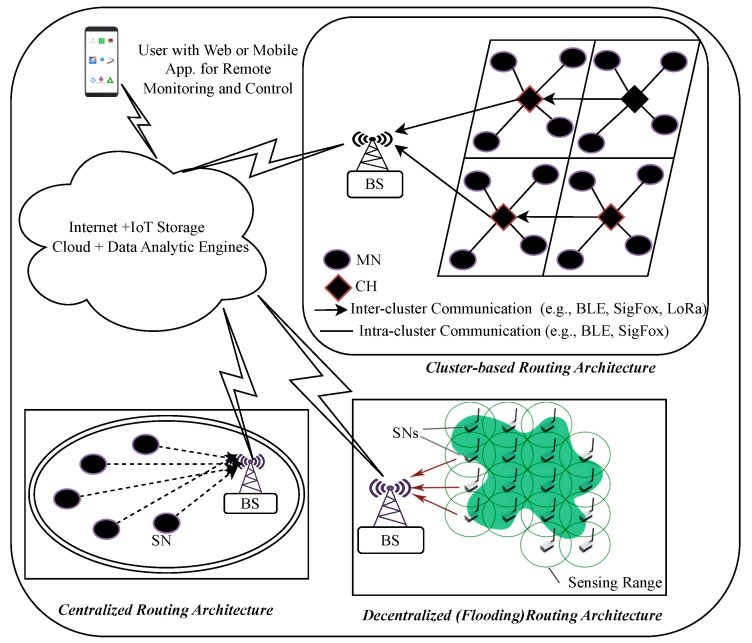
Sample network architectures: centralized data-centric, cluster-based, and graph/flooding-based architectural frameworks [10,50].

The cluster-based multihop architecture is particularly well suited to address these challenges. Unlike centralized architectures, which are prone to bottlenecks and inefficiencies in larger networks, and decentralized flood-based architectures, which still face some centralized issues, the cluster-based approach organizes SNs into clusters with an elected cluster head (CH). This arrangement reduces intra-cluster communication distances, improves energy efficiency, and enhances fault tolerance by distributing the communication load and managing data locally before transmission [4,8,17,38]. Therefore, the cluster-based multihop architecture is ideal for creating robust, cost-effective, and easy-to-manage Agri-IoT networks that can be deployed and operated effectively in agricultural environments.

To date, Agri-IoT technology has struggled to achieve its transformative potential in agriculture due to several factors. These include the reliance on conventional IoT designs that do not fit agricultural settings, the lack of comprehensive context-based design synthesis, underestimation of Agri-IoT’s impact on food security and agricultural productivity, and dependence on electricity and cellular technologies that limit usability in resource-poor regions [3,4,19,35,50]. Addressing these challenges through targeted research and context-specific solutions is essential for advancing Agri-IoT technology.

### 2.2. Synthesis of Related Literature

Within the evolving landscape of agriculture, Agri-IoT represents a transformative paradigm designed to address various field-specific challenges through diverse technological solutions. By leveraging embedded WSNs and actuators, Agri-IoT facilitates the collection and management of farm data, thereby optimizing farming processes such as resource management (e.g., water and fertilizer optimization), irrigation control, and soil management. This technology enables farmers to remotely monitor and manage their crops and livestock using smartphones, potentially reducing production costs and enhancing productivity on limited land. However, the combined effects of climate change and rapid global population growth impose significant constraints on resources such as land, water, nutrients, and labor, necessitating improved resource management and waste reduction.

In response to these challenges, the United Nations has urged researchers to focus on developing contextually relevant Agri-IoT solutions capable of increasing food production by 70% by 2050 to meet the demands of a growing population [20,21]. Nonetheless, current research has predominantly centered on location-constrained and costly testbeds, which are primarily effective under indoor conditions. The following challenges have been identified in the existing literature:**Unique Agricultural Settings:** The agricultural setting is a unique area where conventional IoT technologies do not apply. For instance, existing Agri-IoT solutions are location-constrained because they are based on costly Wi-Fi or cellular communication technologies that require reliable on-farm electricity and internet coverage, which are generally not available in Africa. This necessitates a solution based on affordable battery-powered SNs. These Agri-IoT applications mainly utilize architecture-restricted, high-resource-demanding routing techniques (e.g., routing over low-power and lossy networks protocol (RPL)) and communication standards (e.g., 4G, 5G, ZigBee, NB-IoT, Wi-Fi, and long-term evolution (LTE)) [51], which are difficult to access on typical African farms. Additionally, the users of Agri-IoT technologies are low-income earners with limited technological expertise. As a result, there is a pressing need for contextually relevant, affordable Agri-IoT solutions that are easy to deploy and manage, battery-powered, and leverage freely available technologies without licensing requirements. This study addresses these challenges by focusing on low-power communication technologies, power optimization, and cost-effective, scalable solutions [4,8,17].**Susceptibility to Faults and Failures:** Agri-IoT networks are prone to faults and failures due to the dense deployment of resource-constrained SNs in harsh environments, where they operate autonomously with minimal post-deployment maintenance. Effective network supervisory protocols must incorporate power optimization, fault management, and self-adaptability. The multihop cluster-based architecture has been identified as a promising routing protocol for addressing these issues [3,4,8,9,17]. However, the lack of comprehensive, contextually relevant hardware validation for multihop CA-IoT networks results in abrupt failures in many existing testbed solutions [3,5,14,15,16,31].**Inadequate Consideration of Communication Technologies:** Existing Agri-IoT studies often emphasize high-power-demanding communication technologies (e.g., Wi-Fi and cellular technologies), centralized ZigBee standards, and conventional IoT principles [3,5,14,15,19,31]. Although theoretical studies suggest that multihop cluster-based architectures are well-suited for Agri-IoT applications [4,8,17,52], there is a lack of practical hardware implementation and performance evaluation for these architectures. Many current WSN-based Agri-IoT solutions rely on energy-inefficient standards and require extensive infrastructure, leading to high costs and complexity [3,5,14,15,19,31]. An in-depth assessment of how low-power communication standards like LoRa and Bluetooth Low-Energy (BLE) perform in MCA-IoT networks could significantly benefit both the IoT community and farmers.**Underestimated Potential of Agri-IoT:** The role of Agri-IoT in addressing food insecurity, improving crop quality, alleviating poverty, and boosting agricultural production has been insufficiently explored [3,4,35,39,42,53,54]. The MCA-IoT model presents significant opportunities to tackle food and employment insecurity and enhance crop quality and economic conditions for farmers, but these potential benefits have yet to be fully realized due to inadequate research and development efforts.

Table 1 provides a comparative assessment of established Agri-IoT testbeds. The table highlights that while appearing straightforward with a limited number of SNs, the single-hop centralized architecture becomes increasingly complex to manage as the number of SNs grows. Despite its simplicity, this approach has been extensively researched over the past two decades. However, these solutions are not contextually relevant to Africa because of the unique agricultural settings.

## 3. Overview of Multihop CA-IoT Architectures and Testbeds

Agri-IoT technology can automate various farm management processes, such as precision irrigation, chemical application, and remote disease management [4]. This technology ensures remote monitoring, planning, and control of farm processes through battery-powered SNs equipped with sensing (e.g., DHT22 and STEMMA soil moisture sensors), processing (e.g., Raspberry Pi and Arduino Uno), and communication capabilities (e.g.,  LoRa, ZigBee, BLE, Wi-Fi, NB-IoT, SigFox) [39,41,42]. Since these networks are compelled to drive on battery power and are constrained by distance-limited communication technologies, the embedded supervisory routing protocol must adopt a network architecture that supports efficient power optimization, simplified network management, fault tolerance, and self-adaptability [1,3,4]. Within this research area, the MCA-IoT approach has emerged as the most suitable candidate to harness these advantages when implemented in custom-built testbeds [9,10].

In a typical CA-IoT-based network, such as that shown in Figure 6, SNs are grouped into either static or dynamic clusters, each with an optimally selected cluster head (CH) to minimize intra-cluster communication distances among member nodes (MNs). An MN samples the soil moisture/livestock data and transmits it to its CH. A CH aggregates the received readings from its MNs, executes error and data redundancy control checks, and communicates directly or via a relay CH (RCH) to a BS/gateway to increase the event data quality and network lifespan. This approach enhances data quality and network longevity by reducing data inconsistency errors and minimizing the power wasted on transmitting redundant, long-range data to the BS.

The BS can either process the received data and make local actionable decisions to actuate the irrigation system or send the raw/preprocessed data to the IoT cloud for further processing and return the actionable decisions to the BS for execution. The resulting actuation signal is then sent to the pump for precision irrigation or chemical application.

Data copies can be stored in the IoT cloud for remote monitoring/control or further analysis to identify field climate patterns for automating the irrigation system. Since communication is the principal power consumer in WSN-based CA-IoT networks, and its key factors (i.e., distance and packet size) can be optimally managed through a cluster-based architecture [17,55,56,57,58], this architecture is deemed optimal for power optimization in these networks. Despite the advantages of a cluster-based architecture for WSN-based Agri-IoT applications [9], this claim remains unvalidated due to insufficient research on hardware evaluation and the exploration of its untapped potential, especially within the MCA-IoT architecture.

### Theoretical Analysis of Multihop CA-IoT Design and Operation

In MCA-IoT testbeds, the communication technology, event routing architecture, and sensor node (SN) power supply are key factors influencing operational stability, cost, and deployment management complexity [4,8]. Due to the high cost and limited availability of Wi-Fi and cellular communication technologies, low-power wireless technologies such as LoRa, BLE, ZigBee, IEEE 802.15.4, Z-Wave, and SigFox are preferable. These technologies, grounded in suitable routing topologies and theoretical frameworks [4,8,17], offer a combination of affordability [16,31], robustness, and simplicity [4,31], making them ideal for widespread adoption. For example, ZigBee and Z-Wave, with ranges from 10 to 100 m, are designed for low-power applications and are most effective when deployed in low-density, mesh-like, or centralized topologies [19]. They are typically used in home automation and operate under line-of-sight conditions depending on environmental factors. However, they are characterized by high latency, unreliable Medium Access Control (MAC) protocols (CSMA/CA), low data rates, and high susceptibility to interference and multipath fading. ZigBee IP incorporates a 6LoWPAN adaptation layer, an IPv6 network layer, and the resource-intensive RPL routing protocol [9]. Conversely, Wi-Fi, GPRS, and cellular technologies (e.g., LTE-M and NB-IoT) are associated with high power consumption and require specific location or architectural access. Near-field communication (NFC) and Radio Frequency Identification (RFID) protocols, used for very short-range communication (up to 4 cm), are primarily employed in check-in and inventory systems. LoRa and SigFox, while supporting long-range communication, are limited by fixed network sizes, low data rates, and message capacities [19]. BLE, being a short-range protocol, supports clustering architectures with significant energy-saving potential. However, the critical issues related to these architectures, especially environmental constraints, have led to network management challenges and abrupt SN failures due to rapid battery depletion in previous benchmark testbeds [3,5,14,15]. To optimize the performance of an MCA-IoT network, it is essential to integrate mechanisms that reduce power depletion during sensing, processing, and communication tasks. Given the susceptibility of centralized CA-IoT networks to signal attenuation and retransmissions in dense and obstructive environments, this study recommends using an energy-efficient multihop routing framework, as proposed by the authors of [8], to guide outdoor deployment. Specifically, an optimal approach involves utilizing BLE technology for all intra- and inter-cluster multihop communications while leveraging LoRa for the final relay CH-to-BS communication to achieve comprehensive farm coverage.

Therefore, the proposed multihop route-actuating framework is derived based on the comparative energy efficiency analysis using the $E_{TER,min}$ metric [8]. This metric incorporates the total energy $E_{i,j}$ expended per *L* bits of each i−j route in the single-hop and multihop paradigms. According to Equation (Equation 1), $E_{ij}$ encompasses the distance-dependent transmit power $P_{Tx}$, the power consumed by the transmitter circuit $P_{Tx-elec}$, and the power consumed in the receiver circuit $P_{Rx-elec}$:(1)$$E_{ij}=\frac{\left(P_{Tx}+P_{Tx-elec}+P_{Rx-elec}\right)}{L-bits}$$

$P_{Tx-elec}$ includes the sum of the power consumed by the digital-to-analog converter $P_{DAC}$, mixer $P_{mix}$, transmit filters $P_{fil}$, and frequency synthesizer $P_{syn}$. Similarly, $P_{Rx-elec}$ comprises the sum of the power expended by the frequency synthesizer $P_{syn}$, low-noise amplifier $P_{LNA}$, mixer $P_{mix}$, intermediate frequency amplifier $P_{IFA}$, receive filter $P_{fil}$, and analog-to-digital converter $P_{ADC}$.

Using the free-space PL model $PL\approx {\left(\frac{1}{d}\right)}^{\alpha }$ [56], the received signal’s power $P_{r}=P_{Rx-elec}$ at distance *d* from the transmitter can be expressed as follows:(2)$$P_{r}=P_{Rx-elec}=P_{0}\times {\left(\frac{d_{0}}{d}\right)}^{\alpha },$$
where $P_{0}$ is the known power of the signal received at distance $d_{0}$ and α (2≤α≤5) is the PL exponent.

Given that $P_{Rx-elec}$ is notably significant in practice (approximately 29% of $P_{Tx}$ [59]), a minimum distance between the CH and RCH exists where the single-hop approach becomes more energy-efficient compared to the multihop scheme, and vice versa. Therefore, using the $E_{TER,min}$ metric for any given ER instance and Figure 7, one can compare the energy consumption of single-hop and two-hop transmissions. A radio channel between a transmitter and receiver is established if the received signal strength exceeds the receiver’s sensitivity threshold level ($P_{m}$). Consequently, the CH and RCH must transmit with a minimum power $P_{m}$, defined as
(3)$$P_{m}=P_{x}\times {\left(\frac{d_{0}}{x}\right)}^{\alpha }=P_{y}\times {\left(\frac{d_{0}}{y}\right)}^{\alpha }=P_{z}\times {\left(\frac{d_{0}}{z}\right)}^{\alpha },$$
where *z* is the single-hop distance and *x* and *y* are the two-hop distances such that z=x+y (Figure 7).

If $P_{r}=P_{Rx-elec}$ at the RCH, then two-hop routing will be more energy-efficient if $P_{x}+P_{y}+P_{r}<P_{z}$ is true. Thus,
(4)$$P_{m}\times {\left(\frac{x}{d_{0}}\right)}^{\alpha }+P_{m}\times {\left(\frac{y}{d_{0}}\right)}^{\alpha }+P_{r}<P_{m}\times {\left(\frac{z}{d_{0}}\right)}^{\alpha }.$$

Since $P_{Rx-elec}>0$, there are intervals of distance *x* where multihop communication is disadvantageous compared to single-hop. Consequently, the power saved $P_{saved}$ by the two-hop transmission can be expressed as
(5)$$P_{saved}=P_{z}-\left(P_{x}+P_{y}+P_{r}\right)=\frac{P_{m}}{\beta }\left[z^{\alpha }-\left(x^{\alpha }+y^{\alpha }+t^{\alpha }\right)\right],$$
where β is a constant specified in the model definition. Assuming any potential value of α∈2,3,4,5, say α=5 and $y=z-x,P_{saved}$ and $\Delta P_{saved}$ can be expressed as
(6)$$P_{saved}=\frac{P_{m}}{\beta }\left[z^{5}-\left(x^{5}+y^{5}+t^{5}\right)\right],$$
(7)$$\frac{\delta P_{saved}}{\delta x}=\frac{P_{m}}{\beta }\left[5z^{4}-20xz^{3}+30z^{2}x^{2}-20zx^{3}\right].$$

Given that ϵ=x/z, as derived in Equation (Equation 7), the optimal inter-CH distance ratio is ϵ=1/2. This indicates that the power saved $P_{saved}$ by two-hop routing is maximized, or multihop routing is more energy-efficient, when the RCH is equidistant from both the CH and the BS. To minimize energy consumption in the network, our proposed MCA-IoT approach makes routing decisions based on this condition to ensure $E_{TER,min}$.
(8)$$\frac{\delta P_{saved}}{\delta x}=\frac{P_{m}}{\beta }5z^{4}\left[\left(\epsilon -\frac{1}{2}\right)\left(-4\epsilon^{2}+4\epsilon -2\right)\right]=0.$$

With the aid of Figure 8 and applying the established equidistant multihop actuating principle from Equations (Equation 7) and (Equation 8), the $P_{m}$ for an |n|-hop MCA-IoT network with static SNs can be expressed as
(9)$$P_{m}=P_{1}{\left(\frac{d_{0}}{d}\right)}^{\alpha }=P_{2}{\left(\frac{d_{0}}{d/2}\right)}^{\alpha }\ldots P_{n}{\left(\frac{d_{0}}{d/n}\right)}^{\alpha }.$$

The relationship between $P_{n-hops}$ and $P_{1-hop}$ is also deduced from Equation (Equation 9) via the following equation:$$P_{m}=P_{1}{\left(\frac{d_{0}}{d}\right)}^{\alpha }=P_{2}{\left(\frac{d_{0}}{d/2}\right)}^{\alpha }=P_{3}{\left(\frac{d_{0}}{d/3}\right)}^{\alpha }$$
(10)$$P_{1}=2^{\alpha }P_{2}=3^{\alpha }P_{3}=n^{\alpha }P_{n},$$
(11)$$P_{1-hop}=P_{1},P_{2-hops}=P_{2}+P_{2}=2\frac{P_{1}}{2^{\alpha }}\ldots P_{n-hops}=n\frac{P_{1}}{n^{\alpha }}.$$

It is evident from Equation (Equation 11) that multihop ER will be more energy-efficient than single-hop transmission if α>1 and $P_{Rx-elec}$ is excluded, which validates our implementation results in [8]. However, $P_{Rx-elec}$ cannot be ignored since it is significant in many practical IoT communication platforms. For our MCA-IoT network, $P_{n-hops}$ with $P_{Rx-elec}$ can be expressed as
(12)$$P_{n-hops}=n\frac{P_{1}}{n^{\alpha }}+\left(n-1\right)P_{Rx-elec}.$$

From Equations (Equation 11) and (Equation 12), it can be concluded that multihop routing is more energy-efficient than single-hop routing only if the receiver’s power consumption obeys Equation (Equation 13). The $P_{Rx-elec}$ values from the transceiver platforms are approximately the same as the $P_{1-hop}$ values, which makes them unsuitable for multihop CA-IoT networks. However, recent communication standards such as LoRa, SIGFOX, BLE, and NB-IoT can support multihop routing since the $P_{Rx-elec}$ values are significantly less than the following:(13)$$P_{Rx-elec}<\frac{\left(n^{\alpha -1}-1\right)P_{1}}{\left(n-1\right)n^{\alpha -1}}.$$

From the above expressions, energy-efficient multihop routing in large-scale CA-IoT networks can be achieved under the following conditions:The ratio of multihop distances between the MN, CH, RCH, and edge RCH must be unity along the path of a distant CH (Equation (Equation 8)).To ensure the desired energy savings in multihop routing, α≥2 must be met, and the condition in Equation (Equation 13) must be satisfied by the transceiver module. Specifically, $P_{Rx-elec}$ of the RCH must be sufficiently small compared to $P_{Tx}$ in single-hop routing, as observed in LoRa, SIGFOX, and BLE systems.To further minimize $P_{Rx-elec}$ due to extended durations in multihop routing, a novel routing technique integrating the timeslot assignments of a distant CH into the RCH’s TDMA schedule for its MNs per duty cycle, as proposed in [17], is adopted.

## 4. Proposed MCA-IoT Design

In this section, we provide a comprehensive overview of the development process for the proposed MCA-IoT technology, detailing the justification for the hardware components, the software development approach, and their evolution into the final solution for precision irrigation. The selection and assembly of the hardware components, software development, and MCA-IoT operations were guided by the context-relevant metrics presented in Figure 9.

### 4.1. Hardware Component Selection with Justification

The primary hardware components for network device construction are detailed in Table 2 and Figure 2c. For our MCA-IoT network, designed for deployment in diverse locations and harsh environmental conditions in Africa, the Raspberry Pi 3B+ (RPi 3B+) was deemed optimal. It offers robust operational stability under challenging conditions and features a built-in radio module supporting BLE 4.2 and dual-band WLANs (2.4 GHz and 5 GHz). With a 1.4 GHz processor, 1 GB SDRAM, a throughput of 300 Mbps, and a Power over Ethernet (PoE) port via a separate PoE HAT, the RPi 3B+ enables simplified cabling and remote access through a VNC server and Wi-Fi. Among various technologies, BLE 4.2 was selected for intra-cluster and inter-cluster communications due to its energy efficiency, compatibility with mobile units, support for cluster-based routing, and high throughput. Additionally, LoRa was chosen for edge CH-to-BS and BS-to-cloud communications due to similar advantageous characteristics. The technology selection justification matrix and power requirement metrics outlined in [10,50] were utilized for this study. Further details on the selected hardware components are provided in Table 2.

The devices in the MCA-IoT network, including the MNs, CH, RCH, and BS, were custom-built using RPi 3B+ controllers and various cross-platform hardware and software components tailored to their specific roles and demands, as shown in Figure 10. These devices are scalable and capable of role rotation. The MNs collect soil temperature and moisture data, which are transmitted to the CH. The CH processes the data by removing outliers, aggregating it, and then sending it directly or via an RCH to the BS for local decisions, while retaining a copy in The Things Network (TTN). Data from distant CHs are routed through BLE-based multihop for inter-cluster communications, while edge CHs (ECHs) use LoRa technology for data transmission to the BS. The two ECHs are equipped with LoRa end nodes to facilitate long-range communication. An actuation signal triggers the irrigation system to activate the water pump. The RPi 3B+ was chosen for its stability across various environmental conditions. The hardware and software design of the MCA-IoT was guided by the following considerations:**Architectural Support, Operational Simplicity, and Affordability:** Cost-effective and easy-to-operate components and technologies (e.g., BLE, LoRa, RPi 3B+) were selected to leverage the multihop cluster-based architecture. This choice ensures that the MCA-IoT network is infrastructure-less, battery-driven, deployable, manageable in any location, labor-saving, flexible, and easy to integrate into farm environments without fixed infrastructure.**Effectiveness and Dependability:** Components were chosen based on their availability, ease of assembly, and market presence, which supports system maintenance and scalability.**Energy Efficiency:** Low-power components and energy-saving techniques, such as duty-cycled sampling, role rotation, and control of redundant data transmission, were implemented to ensure the network’s efficiency throughout the crop season, especially given the lack of reliable electricity in many SSA farms.**Environmental Robustness:** Hardware was tested under extreme climatic conditions (40 °C to 51 °C), similar to those in the Sahel Region, with the RPi 3B+ performing stably compared to other controllers that failed even under indoor conditions.**Standardized Manufacturing:** The selection of components ensured compatibility and smooth integration, streamlining production and assembly, and maintaining system coherence.

In summary, the effectiveness of the MCA-IoT network is closely linked to the choice of components and the architecture of event routing, ensuring that the desired performance outcomes are achieved.

### 4.2. MCA-IoT Supervisory Software Development Phases and Implementation

Following the hardware assembly, an iterative lean technology development approach, as illustrated in Figure 11, was employed. This method involved analyzing the system’s expected performance metrics, developing custom-built prototypes, evaluating their performance against desired metrics, and iteratively refining the prototypes.

The proposed event-routing protocol incorporates a cluster-based architecture, duty cycle, and fail-over scheduling for event sampling, data aggregation, and data transmission/reception.

This protocol was developed using Python 3.7 within the Raspbian OS of the Raspberry Pi 3 B+. The design and implementation involved cluster formation processes, fault tolerance schemes from [10,17], and cluster quality evaluation methods from [18]. Additionally, BLE 4.2 libraries and drivers for the ATSAMD10 Adafruit STEMMA soil sensor (i.e., I2C Capacitive Moisture Sensor), and DHT22 sensor were installed from Bluez and Adafruit, respectively. The software configuration followed the SN design steps from [10,50], with the Bluepy module installed for handling BLE device communications using code from the BlueZ project. The main software running on the BS, CHs, and MNs, as well as their modus operandi, as outlined in Figure 12, are detailed as follows:The proposed MCA-IoT network utilizes three sets of distributed network management and sampling software: MN/client software, CH client/server software, and BS/gateway/server software. The operational steps and roles of the software programs for intra- and inter-cluster communications are illustrated in Figure 12. The MN software locally hosts a seesaw-embedded Bluetooth socket client in a Python 3.7 script for managing network construction, event sensing, BLE transceiver operational state moderation, and transmission of sampled data to the BS at specific crontab timeslots. This intra-cluster software, shown in Figure 12a, was designed using Bluetooth socket programming, the seesaw module, the BlueZ module, local time, built-in crontab -e in Python 3.7, and NTP from time.google.com.A CH incorporates client and server codes within a single software package, enabling it to receive data from MNs (server) and distant CHs and to act as a client to the BS or RCHs. As shown in Figure 12b, this software also incorporates fail-over tasks and the data aggregation mechanism illustrated in Algorithm 1. Depending on the physical location of the CH (i.e., if the CH is not an edge CH to the BS), the fail-over mechanism allows this distant CH to multihop its data via a secondary RCH to the BS if the primary RCH is out of service. All CHs, except the edge CHs, incorporate a fail-over mechanism. Figure 13 illustrates an instance of fail-over, where a secondary RCH routes aggregated data to the BS.This software consists of a Bluetooth socket server and client in Python 3.7 scripts that control cluster construction/reconstruction, as outlined in Algorithm 2; fail-over route actions, as described in Algorithm 3; and packet reception from MNs. It also handles data aggregation with fault tolerance techniques inherited from [17], packet transmission to the next-hop RCH or BS, and management of BLE transceiver states. Thus, a distant CH locally stores raw data from its MN and appends its aggregated data to the next-hop primary RCH or secondary RCH in case of fail-over. Unlike a distant CH, an RCH forwards the aggregated data received from its MNs and appends the aggregated data from distant CHs to the next-hop RCH or edge CH. An edge CH receives data via BLE and forwards the locally aggregated data from its MNs; the appended aggregated data from the distant CH/RCH is appended to the BS via LoRa technology. Figure 12b,c illustrate these tasks. The proposed software adapts to scalable network conditions caused by SN out-of-service faults and variations in SN count or network size. The CH software utilizes modules such as Bocket, Bluetooth, sys, threading, time, csv, Queue, struct, ntplib, gspread SCL, and SDA from board, busio, and seesaw from adafruit_seesaw.seesaw .Similarly, the BS software also contains the Bluetooth socket server script in Python 3.7 for managing network construction, reception and storage of event data, BLE transceiver operational state moderations, and transmission of received sampled data to the cloud at specific timeslots. This part of the software running on the BS uses Bluepy, BlueZ, and APIs to communicate with edge CHs via the BLE module at scheduled times to receive and store sampled data locally in a .csv file. Additionally, this software incorporates LoRa communication technology to transmit the received farm data to the TTN cloud for remote monitoring. The LoRa communication technology was configured to have a transmitter power output of 14 dBm, a spread factor of 7, a bandwidth of 125 kHz, and a coding rate of 4/5. The BS further processes the received data, and if the result exceeds a preset threshold, an actuation signal is sent to the pump to start the irrigation system. This software also accounts for SN out-of-service faults from the CHs by indicating the MN and CH counts of the received aggregated data. The main modules imported into the BS software include socket, Bluetooth, sys, threading, time, csv, Queue, struct, ntplib, and gspread, whereas the MN relies on Bluetooth, time, SCL, and SDA from board, busio, and seesaw from adafruit_seesaw.seesaw. The sensory data are the main parameters transmitted at the crontab-scheduled timeslot to the BS and TTN cloud at 3-h intervals to reduce data noise while capturing significant changes in soil parameters.
**Algorithm 1** Data aggregation in the sensor network**Require:** n≥0**Ensure:** $y=x^{n}$1:Each MN in the cluster samples or senses data during the scheduled sensing period.2:Each MN sends its data to the selected or dedicated CH in that sensing cluster.3:The CH polls the received data and checks if all the member nodes have sent their data.4:The CH adds/aggregates received MNs into one.5:$\mathrm{Aggregate}\text{\_}\mathrm{Data}=\sum_{i=1}^{N}{\mathrm{Node}\text{\_}\mathrm{Data}}_{i}$▹ Where *N* is the total number of member nodes6:The CH checks for redundancy in the aggregated data to avoid false positives.7:The CH represents the sensing cluster’s area measurement in the aggregated data.8:y←19:X←x10:N←n11:**while** N≠0 **do**12:    **if** *N* is even **then**13:        X←X×X14:        $N\leftarrow \frac{N}{2}$                                                                                          ▹This is a comment15:    **else if** *N* is odd **then**16:        y←y×X17:        N←N−118:    **end if**19:**end while** 

The BS processes aggregated data in .csv format and appends a copy to the TTN cloud at scheduled intervals using LoRa communication technology.

To assess the efficacy of the proposed system, it was tested under various environmental conditions, both indoors and outdoors. Modifications were made to the architecture during each deployment, and performance impacts were monitored and analyzed using MATLAB R2022b, OriginStudio 2022, and WebplotDigitizer version5.

A total of four clusters, each comprising four MNs, were tested. The assembly of network components and circuitry for a cluster is shown in Figure 10, with indoor and outdoor testbed setups illustrated in Figure 14a,b. Compared to existing Agri-IoT solutions, the MCA-IoT system offers easier deployment and management, is more cost-effective, energy-efficient, simpler, fault-tolerant, and less location-dependent than the state-of-the-art solutions shown in Figure 2b.
**Algorithm 2** Bluetooth sensor network setup1:**Intra-Cluster Setup:** 2:**for** each cluster *i* in set *C* **do**3:    $\mathrm{pair}(CH_{i},N_{i})$4:    $\mathrm{trust}(CH_{i},CH_{i})$                             ▹ since a cluster head is a master in its own cluster5:    $\mathrm{secure}(CH_{i},N_{i})$6:**end for**7:** ** 8:**Inter-Cluster Setup:** 9:**for** each initiating cluster head $CH_{i}$ in set *C* **do**10:    $\mathrm{trust}(CH_{i},CH_{i+1})$                                                 ▹ where $CH_{i+1}$ is the next-hop relay11:    $\mathrm{pair}(CH_{i},CH_{i+1})$12:**end for** 13:** ** 14:**Adding New Cluster:** 15:**function**AddNewCluster(new_cluster)16:    Add new_cluster to set *C*17:    Perform intra-cluster setup for new_cluster18:**end function** 

**Algorithm 3** Fail-over algorithm for out-of-service RCH
**Require:** Set *R* of all cluster heads in the sensing network**Ensure:** Data transfer until it reaches the central base or sink station
  1:Let $RT_{CH_{i}}$ be the routing table of cluster head $CH_{i}$, mapping cluster head IDs to MAC addresses and static IP addresses.  2:Let $NH_{CH_{i}}$ be the next-hop cluster head selected by cluster head $CH_{i}$.  3:Let $D_{CH_{i}}$ be the distance between cluster head $CH_{i}$ and the next-hop cluster head.  4:**Initialization**:  5:**for** each cluster head $CH_{i}$ in set *R* **do**  6:    Initialize routing table: $RT_{CH_{i}}=\left\{\left(ID_{j},MAC_{j},IP_{j}\right)\;|\;\mathrm{for}\;\mathrm{all}\;j\right\}$  7:**end for**   8:** **   9:**function** HopData($CH_{i}$)  10:    Check routing table: $NH_{CH_{i}}=RT_{CH_{i}}\left(ID_{NH_{i}}\right)$  11:    **if** $NH_{CH_{i}}$ is down or out of service **then**  12:        Select next-hop cluster head with same distance: $NH_{CH_{i}}=\arg\min_{j}\left\{D_{CH_{i}},\;s.t.\;RT_{CH_{i}}\left(ID_{j}\right)\mathrm{is}\;\mathrm{available}\right\}$  13:        Trigger cron job to set up Bluetooth connection: $\mathrm{SetupBluetooth}(CH_{i},NH_{CH_{i}})$  14:    **end if**  15:    Transfer data to $NH_{CH_{i}}$  16:**end function**   17:** **   18:The process continues until the data hit the central base or sink station.


## 5. Experimental Setup, Results, and Discussion

The MCA-IoT technology was deployed and evaluated with a six-cluster configuration, as depicted in Figure 10, under both indoor and on-farm environments, demonstrating scalability. BLE-based intra- and inter-cluster distances were determined to be 5 m using the theoretical approach outlined in Section 3, Section 4 and Section 5, following the methodology proposed by the authors of [10] to mitigate potential packet drops. The LoRa-based communication distances measured were 2.1 km for line-of-sight (LoS) and 1.5 km for non-line-of-sight (NLoS) scenarios. However, since the LoRa base station was deployed on the farm, these distances were considered insignificant for our experimental setup.

### 5.1. Experimental Setup

The on-farm deployment schematic, which incorporates equidistant inter- and intra-cluster placements based on the power optimization framework described in Section 3, is illustrated in Figure 15. While the distance remains constant, BLE technology performance may vary due to environmental factors such as physical obstructions, dust concentration, and interference from other wireless technologies like Wi-Fi and ZigBee [10,50]. In our case, a 5-m distance was chosen, aligning with the threshold range for stable BLE connectivity and minimal interference observed in Tarkwa, Ghana, where the range was between 5 and 6 m. This selection is consistent with the sensing coverage requirements for agricultural soil proposed in [19,31]. The 5-m distance ensures optimal range overlap, comprehensive farm coverage, and reliable connectivity, particularly when battery power falls below 30% of its rated capacity. The setup and deployment of the MCA-IoT network followed a four-step process:**Deployment of Static Nodes:** Static nodes were positioned based on the cluster quality estimation technique from [18]. BLE radio RSSI levels were estimated, and all network components were powered by solar-rechargeable power banks. These were installed on stands elevated above the crops to prevent shading, ensuring adequate sunlight exposure. Soil moisture sensors were installed in the soil before activating the MCA-IoT network.**Activation and Configuration:** The MCA-IoT network participants were activated, clusters were formed, and a 3-h sampling schedule was established using the onboard crontab-e of the Raspberry Pi 3 B+.**Data Transmission and Processing:** During the experimental phase, the mobile nodes (MNs) transmitted sampled data to the base station (BS) through their respective cluster heads (CHs), relay CHs (RCHs), and edge CHs for processing, decision making, TTN cloud database updates, and irrigation system control relay actuation.**Performance Evaluation:** The network’s performance stability was assessed over a complete farming season lasting 3 months, during which the aforementioned steps were repeated to account for extreme climatic conditions and variable participant activity durations.

The final MCA-IoT network, comprising four MNs per CH, two distant CHs, two RCHs, two ECHs, and one BS, in the two deployment modes, is illustrated in Figure 14.

### 5.2. Results and Discussion

Since all the aforementioned post-deployment performance parameters culminate in a node’s availability or lifespan, this section discusses the key modalities for ensuring energy efficiency. With the aid of the UM25C, Type-C USB 2.0 Full-Color LCD Multimeter Tester, participants’ principal performance parameters, such as energy, current, voltage, power, and ambient temperature in both °C and °F, were remotely monitored and recorded via this Bluetooth-enabled USB dongle placed between the power supply and RPi-based MN, CH, RCH, and edge CH (ECH). The ECH receives aggregated data from the RCH via the BLE and forwards it to the BS via a LoRa communication device. This UM25C dongle comes with PC software that offers a real-time graphical view of these parameters and can extract them into .csv files for further analysis.

The initial deployment and prototype iterations were conducted in the laboratory for a month, as illustrated in Figure 14a. Afterward, the proposed MCA-IoT technology was deployed on the project implementation farm, as shown in Figure 14b. Although the technology was designed to mitigate drought challenges and support year-round farming, this study evaluated performance only during the dry season. The impact of environmental conditions on the network’s performance and potential risk factors remain to be fully understood. Throughout the three-month deployment period, the system continuously transmitted data to the TTN cloud without any network participant failures.

With the aid of our mobile app, as shown in Figure 16, and sample real-time soil temperature and moisture data, as shown in Figure 17, the farmer could view the farm’s conditions and send a GSM-based message to operate the irrigation system, as illustrated in Figure 18a. This ensured that the correct amount of water was applied to the crops at the optimal time. Consequently, the Crop Department of Ghana’s Ministry of Food and Agriculture, Tarkwa Nsuaem Unit, confirmed that our farm produce was of comparatively better quality. Samples of the vegetables harvested from our farm are shown in Figure 18b, and the crop quality is evident.

Unlike related Agri-IoT studies [3,5,14,15,19,25,31,60,61,62,63,64] that only focus on developing workable real-world Agri-IoT solutions using classical IoT design principles and technologies without incorporating real-world performance data sampling and data processing, this study proposed and custom-built a robust, affordable, context-relevant, and globally significant MCA-IoT remedy with prototypes and unique indoor and outdoor performance results. Since the effects of all the critical design metrics (i.e., FM, adaptability, and power consumption optimization) were revealed in the SN lifespan (SN out-of-service) and data outliers [10], of which the latter were auto-addressed by the embedded routing protocol, the power depletion rate or SN lifespan remained our principal real-time performance metric. The power depletion patterns of the MNs, CHs, RCH, and ECH under different outdoor environmental conditions are presented in Figure 19. The LoRa BS is excluded because it was located in the lab where power is reliable. As depicted in Figure 19, the power depletion rate scaled with the network participant’s task. For instance, the MN consumed the least power because it only samples and sends measurements to its CH. A CH receives the data, aggregates it, and forwards it to the next-hop RCH at a crontab-scheduled timeslot. Although the RCH also aggregates its MN data and forwards aggregated data from a distant CH, there was not much appreciable power depletion difference between a CH and an RCH due to the improved efficiency of the scheduling algorithm. However, the ECH showed an appreciable power depletion difference due to the inclusion of the LoRa communication device, which consumed more power than the BLE. None of the network participants’ power was depleted beyond 70% because the 60,000 mAH solar-powered power bank’s recharging efficiency was extremely high. Thus, using similar hardware components, this study in its novelty presents a reference Agri-IoT routing protocol for the MCA-IoT architecture with task-scalable prototypes and a unique performance assessment technique.

## 6. Conclusions

This study provides a comprehensive evaluation of an adaptive multihop cluster-based agricultural IoT (MCA-IoT) network specifically designed for precision farming and greenhouse applications. The system is characterized by its robustness, cost-effectiveness, scalability, infrastructure independence, location versatility, and fault tolerance. Tested in an agricultural environment, the custom-built network utilizes BLE and LoRa communication technologies alongside a clustering topology to improve performance efficiency and data quality. The technology exhibits notable operational simplicity and flexibility, which facilitates network expansion and enhances user accessibility. Initial testing conducted during the dry season confirms the system’s robustness and adaptability. The technology achieved data transmission with zero packet loss. Transmission covered up to 2.1 km using the LoRa technology for LoS and 1.5 km for NLoS, saving transportation costs and potential costs resulting from crop loss. Power depletion for all network participants did not exceed 70% with a 60,000 mAh solar-powered power bank. However, comprehensive validation through extended testing over multiple years and across additional nodes is essential to fully evaluate the system’s commercial viability and identify potential risk factors. Climatic extremes, in particular, may impact the rate of power depletion in nodes, which is a principal risk factor. Limiting testing to the dry season may have restricted the identification of all risk factors associated with adverse weather conditions. Our findings indicate that integrating existing IoT-based communication technologies is the most effective strategy for typical farming contexts in sub-Saharan Africa. Future research should focus on incorporating soil nutrient sensors and remote monitoring technologies to further extend the system’s capabilities and optimize farm resource management.

## Figures and Tables

**Figure 1 sensors-24-06113-f001:**
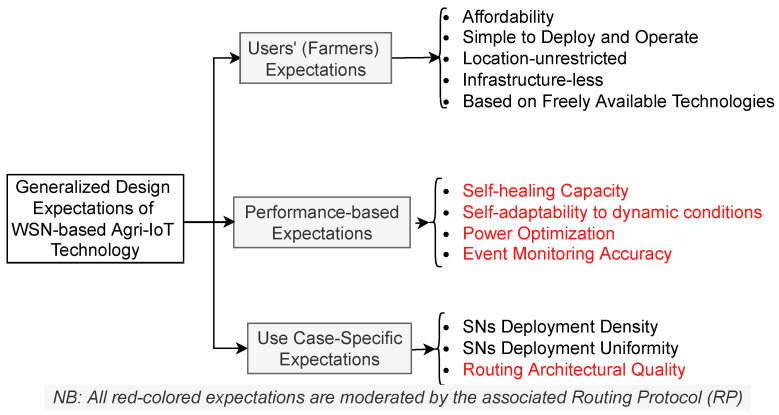
Generalized design expectations of a globally significant Agri-IoT technology [9,10].

**Figure 2 sensors-24-06113-f002:**
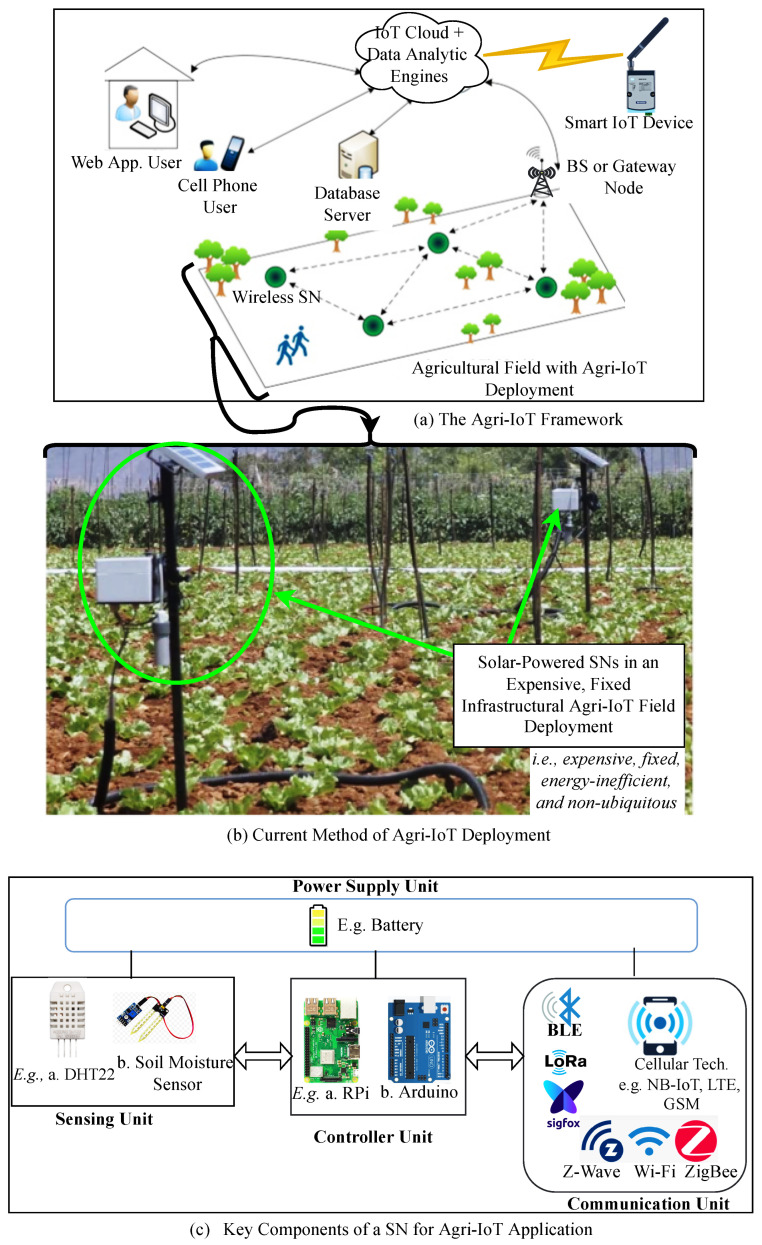
GeneralizedAgri-IoT framework: field layout overview of Agri-IoT framework (**a**), a sample of Agri-IoT in state-of-the-art applications (**b**), and key components of an SN or a BS (**c**) [10].

**Figure 3 sensors-24-06113-f003:**
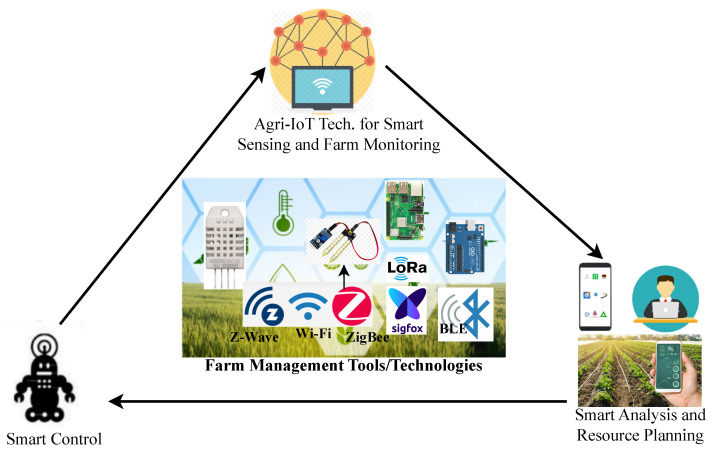
Conceptual framework: Agri-IoT-based farm monitoring and control cycle [9,10].

**Figure 5 sensors-24-06113-f005:**
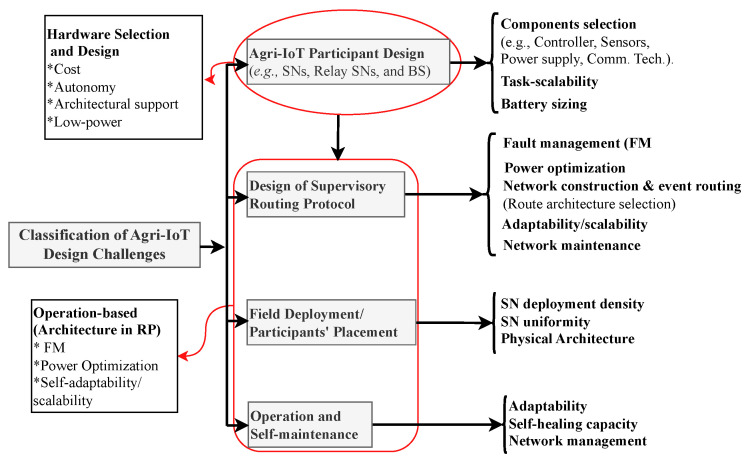
Taxonomy of Agri-IoT design challenges [9,10].

**Figure 6 sensors-24-06113-f006:**
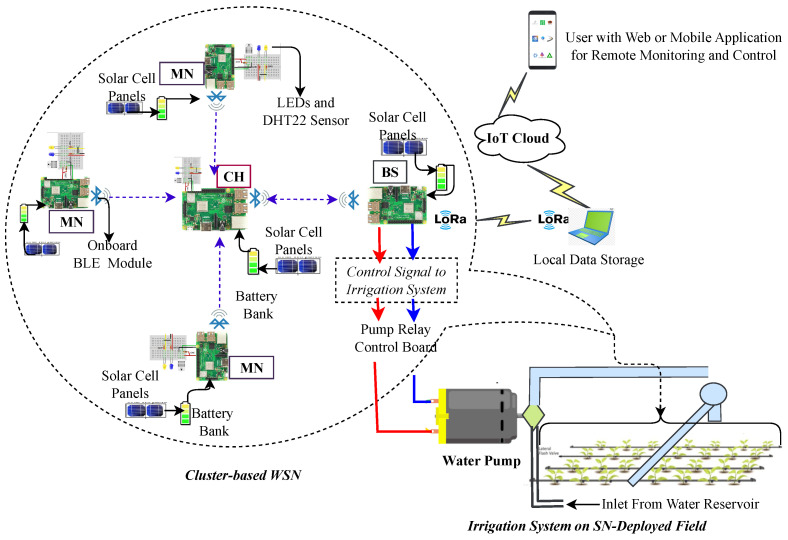
WSN-based Agri-IoT architecture for precision irrigation application [10].

**Figure 7 sensors-24-06113-f007:**
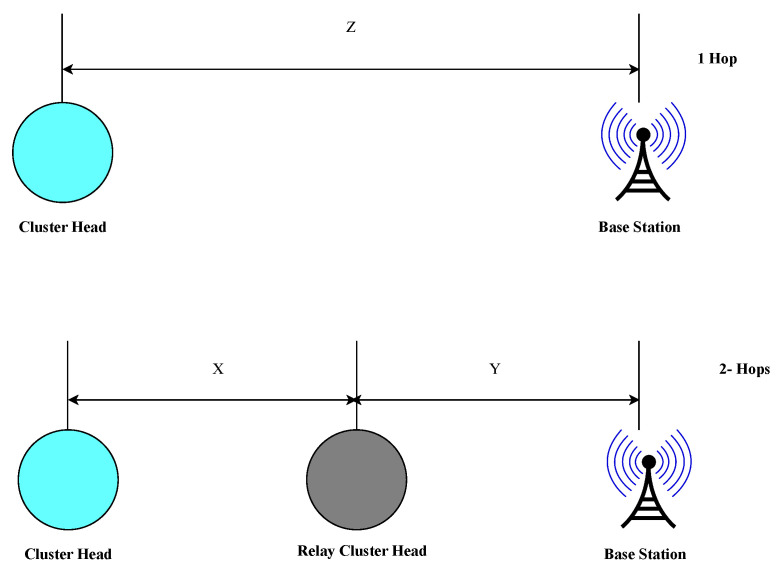
Directhop versus two-hop case.

**Figure 8 sensors-24-06113-f008:**
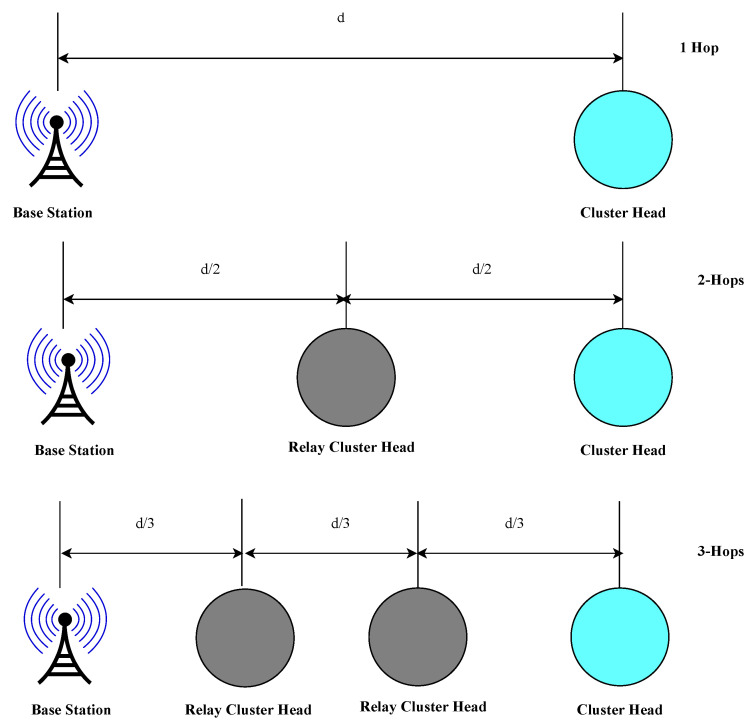
Equidistantmultihop transmission framework.

**Figure 9 sensors-24-06113-f009:**
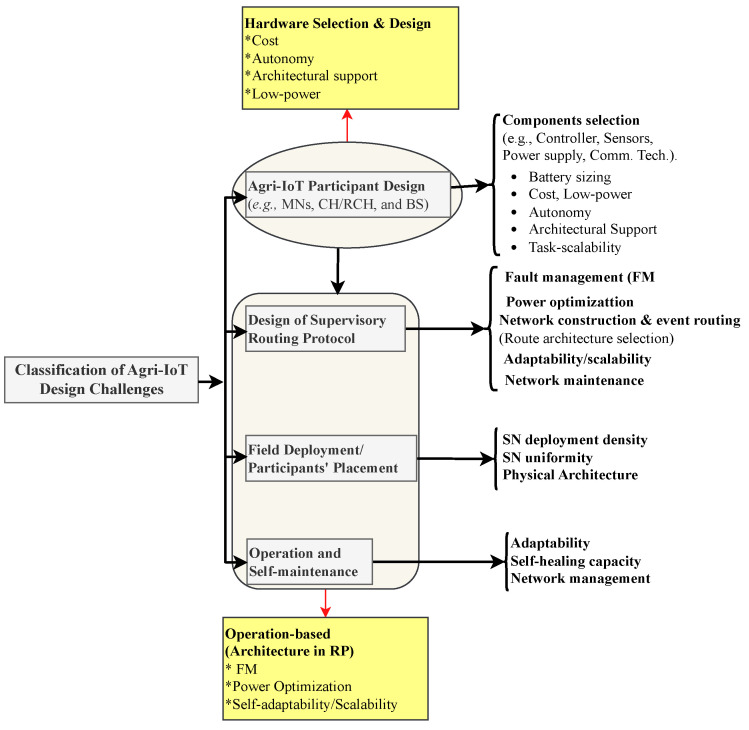
The pivotal guiding metrics for the hardware selection/assembly, supervisory software development, and operation of the proposed MCA-IoT framework.

**Figure 10 sensors-24-06113-f010:**
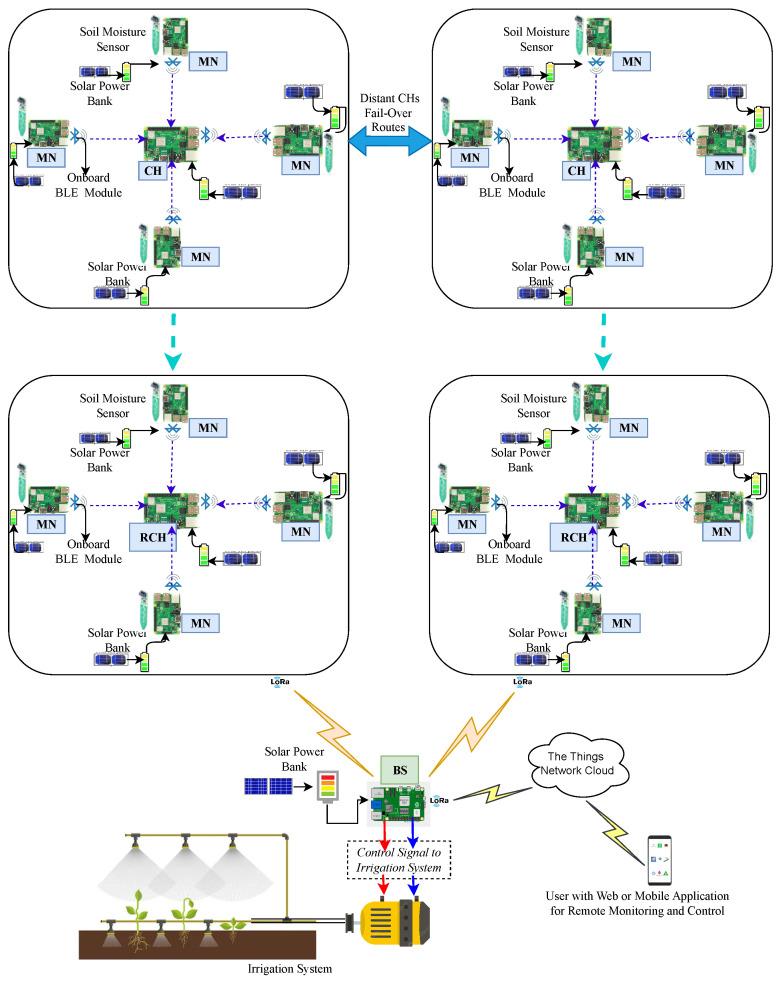
A 4-cluster schematic diagram of the proposed multihop CA-IoT framework.

**Figure 11 sensors-24-06113-f011:**
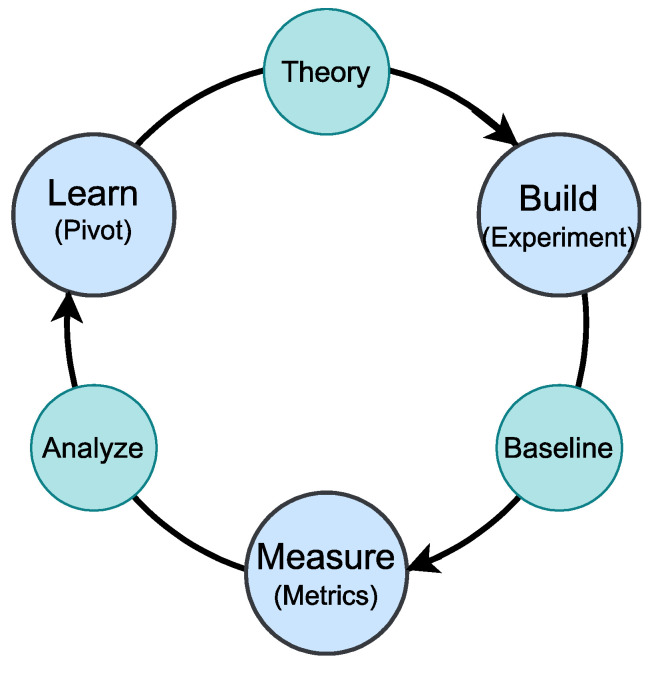
Proposed multihop CA-IoT development method using the iterative lean approach.

**Figure 12 sensors-24-06113-f012:**
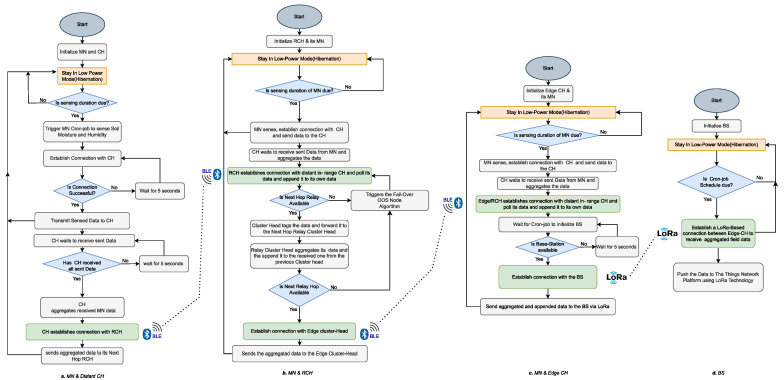
Flowcharts of event-routing operational cycle of software running on intra-cluster and inter-cluster/multihop spaces using MN, CH, RCH/edge CH, and BS devices.

**Figure 13 sensors-24-06113-f013:**
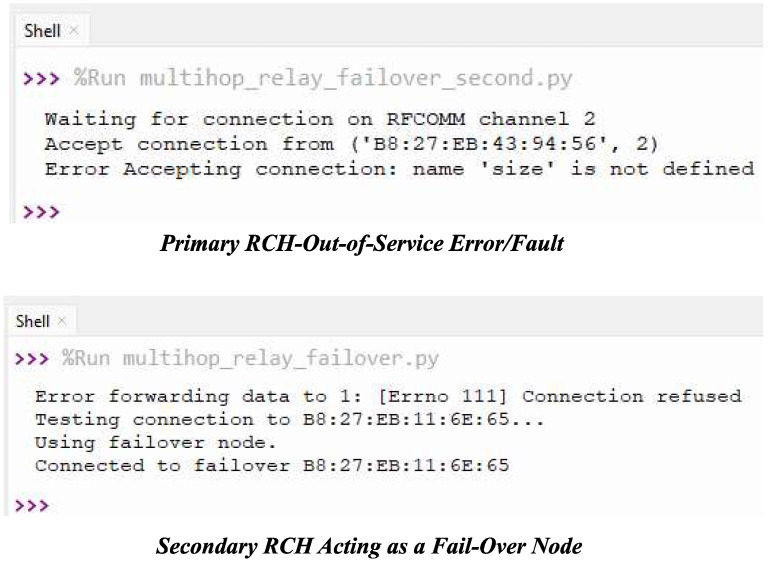
Evidence of implementation of fail-over mechanism.

**Figure 14 sensors-24-06113-f014:**
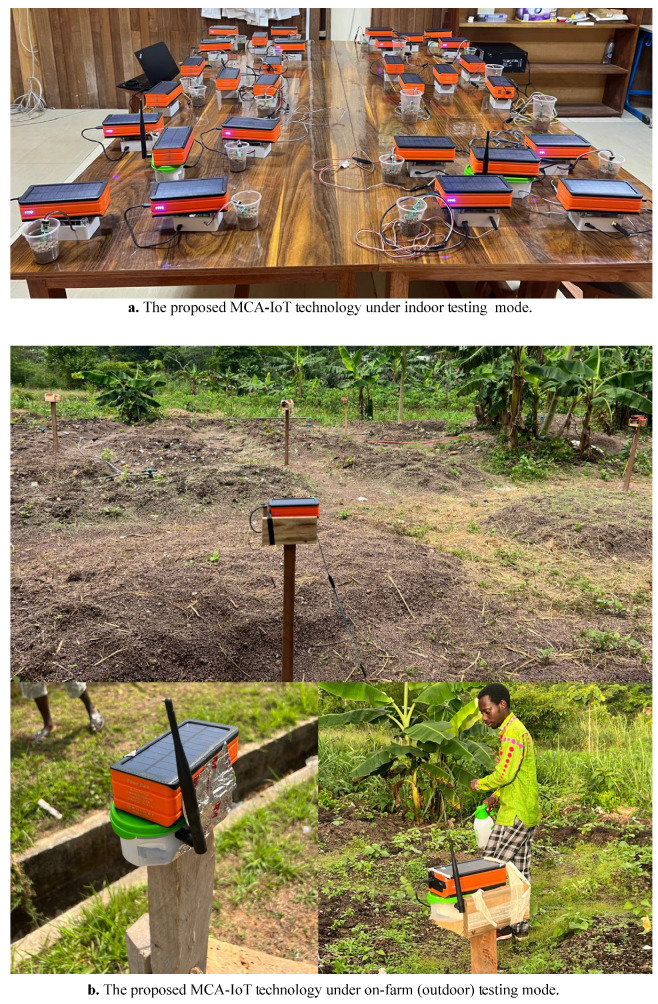
Proposed MCA-IoT technology in full indoor and outdoor operation modes.

**Figure 15 sensors-24-06113-f015:**
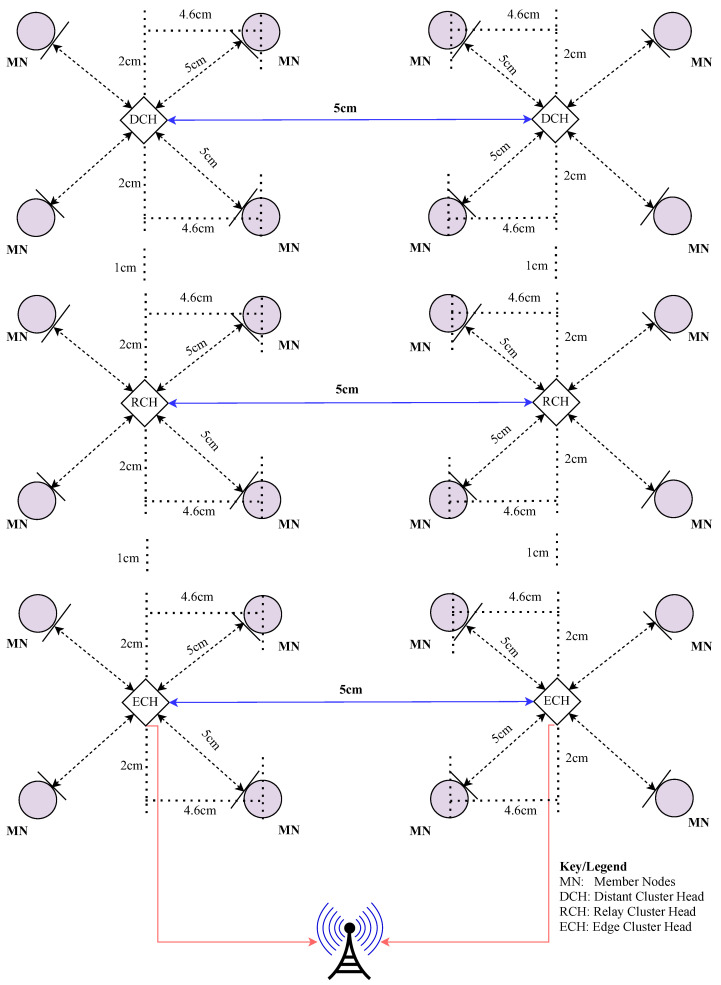
A map showing the experimental setup of the proposed MCA-IoT network.

**Figure 16 sensors-24-06113-f016:**
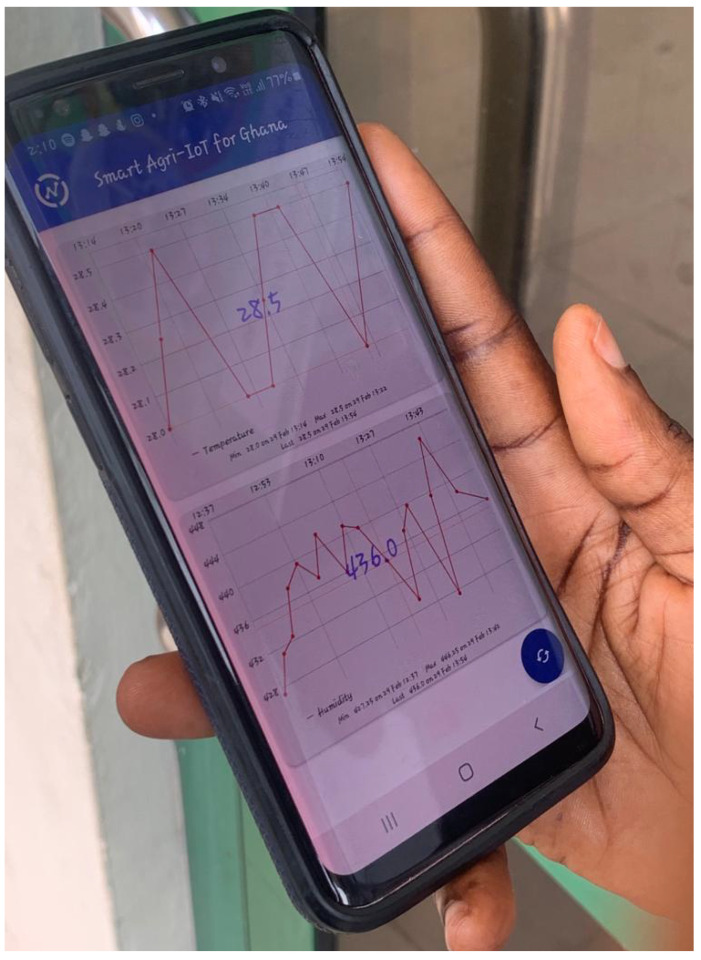
A mobile app showing real-time moisture data from a farm.

**Figure 17 sensors-24-06113-f017:**
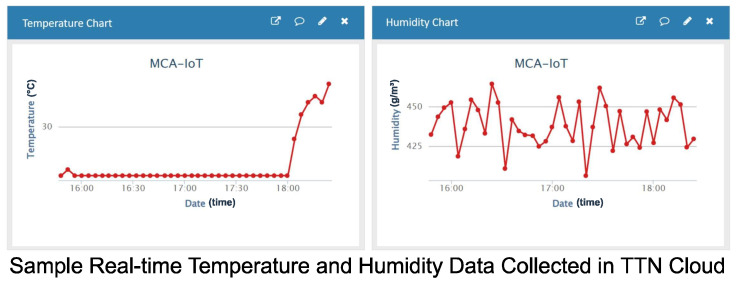
Real-time temperature and humidity data from on-farm deployment, as shown in the TTN cloud.

**Figure 18 sensors-24-06113-f018:**
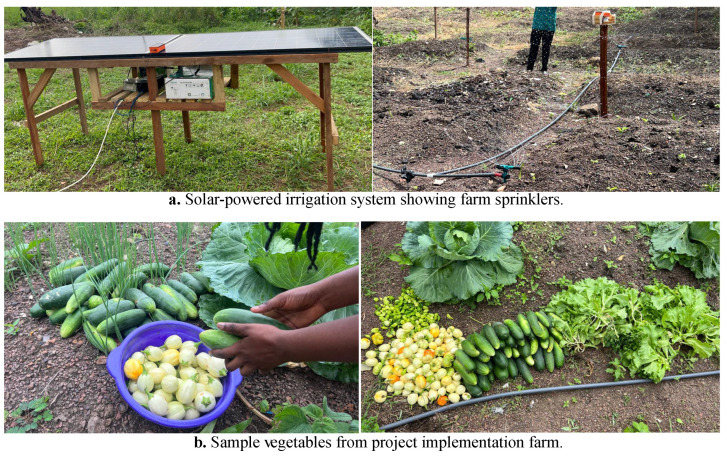
The irrigation setup of the proposed MCA-IoT technology and sample organic farm produce showing high crop quality.

**Figure 19 sensors-24-06113-f019:**
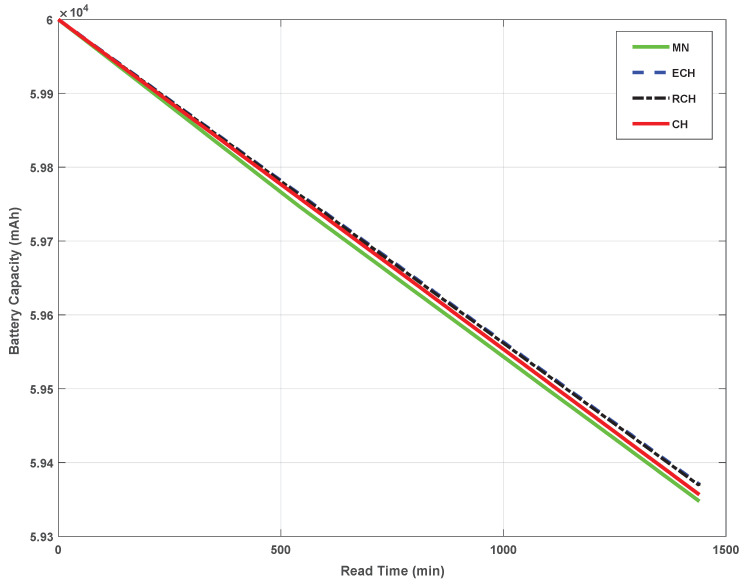
Worst-case power depletion of the proposed MCA-IoT network’s participants.

**Table 1 sensors-24-06113-t001:** Comparative assessment of benchmarking Agri-IoT testbed solutions.

Author/Yr	Comm. Tech	Routing Architecture	Deployment (Indoor/Outdoor)	Hop Type	Problems Addressed	Issues
Ref. [14], 2006	ZigBee	Mica2 clones hardware and TinyOS software/centralized, flooding	Outdoor	Single-hop	Precision farming to gather real-world experiences	Relies on a fixed support system, is expensive, power-inefficient, and location-restricted; no single measurement achieved due to high network complexity
Ref. [3], 2017	ZigBee	IEEE 802.15.4/centralized, flooding	Outdoor	Single-hop	Disease control	Relies on a fixed support system, is expensive, power-inefficient, and location-restricted
Ref. [21], 2017	Wi-Fi	Centralized with one SN via 6LoWPAN	Indoor	Single-hop	Monitors farms with a camera and soil temperature and humidity using CC3200 single-chip	Location-restricted due to Wi-Fi
Ref. [5], 2018	ZigBee	Flooding-based	Indoor	Single-hop	Data outlier detection and decision support system for precision irrigation testbeds	Results based on 3 SNs under unrealistic indoor conditions
Ref. [15], 2018	Wi-Fi-based	6LoWPAN, 6LBR, and centralized, flooding	Indoor	Single-hop	Latency improvement via fog computing	Capital-intensive, energy-inefficient, high-complexity, and location-restricted
Ref. [24], 2018	Arduino Uno with Wi-Fi/3G/4G	Centralized one-SN architecture	Indoor	Single-hop	Monitors environmental conditions	Location-restricted due to Wi-Fi and high complexity when the network scales
Ref. [31], 2019	ZigBee	Centralized, flooding	Indoor and outdoor	Single-hop	Gathers real-world deployment experiences	Results focused on mere observation rather than real-world deployment scenarios
Ref. [28], 2020 Ref. [30], 2021	NodeMCU with Wi-Fi	Centralized one-SN architecture	Indoor	Single-hop	Monitors environmental conditions	Location-restricted due to Wi-Fi and high complexity when the network scales
Ref. [26], 2023	Arduino Uno with GSM SIM900.	Centralized one-SN architecture	Indoor	Single-hop	Monitors environmental conditions	Location-restricted due to Wi-Fi and high complexity when the network scales
Ref. [20], 2024	Esp32 with Wi-Fi and ASR605x/STM32WLE5JC with LoRa	Centralized one-SN architecture	Outdoor	Single-hop	Creates a cheaper IoT SN with easier data storage and processing steps	Location-restricted due to Wi-Fi and high complexity as the network scales

**Table 2 sensors-24-06113-t002:** MCA-IoT hardware components list.

Hardware Component	Function	Why Selected
RPi 3B+ controller	Processing, storage, hosting other hardware peripherals; serves as main controller for MNs, CHs, and BS	Has onboard Wi-Fi and BLE 4.2 radio modules, as well as a PoE port, for easy remote programming; is cheap, ubiquitous, and able to withstand adverse weather conditions
LoRaWAN gateway (BS) with USB-LoRa end nodes	For communication between the edge RCHs and BS, and then the BS and the TTN cloud	Offers cheap and flexible communication options for different MCA-IoT applications
BLE 4.2 Module	For intra-cluster and inter-cluster/multihop communication	Supports the cluster-based architecture without fixed infrastructural requirements; is ubiquitous, simple to implement, and cheap
STEMMA soil moisture sensor (Adafruit)	For sampling soil moisture and temperature data	Is hardy, accurate with a wider/suitable measurement range, and has freely available libraries at the Adafruit Library
Solar-powered 60,000 mAH (5v/3A) LICORNE and ANYFONG power banks	Powers MNs, CHs, RCHs, and BS	Suits network devices’ power requirements

## Data Availability

Data are contained within the article.

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
