# Peer review of "Hardware Development and Evaluation of Multihop Cluster-Based Agricultural IoT Based on Bluetooth Low-Energy and LoRa Communication Technologies"

_sensors, 2024, doi:10.3390/s24186113_

Round 1

Reviewer 1 Report

Comments and Suggestions for Authors

The research presented in this study is very interesting but could be presented better.

It would have helped the reader to have a background section to introduce the different concepts tackled.

- The topic is relevant especially that it is dedicated for Ghana. Also, it is a good idea to have such tools designed for laypeople who have limited technical expertise. The concepts and technologies are not novel but also not consumed, however, their combination and the purpose why they are used makes the work original. This work mainly addresses the lack of hardware studies related to IoT in agriculture. Similar systems have been deployed before but not tested in different conditions.

- This work filled the gap by providing a comprehensive, easy to use, and robust solution to automate farm management.

- The references are relevant and appropriate.

- The tables and images are also good, table 1 might be redone as there a small mistake towards the last row. The figure are not cited in the paper, hence, there is somehow a confusion about each picture and what it means.

The conclusion could elaborate more on the findings.

Overall, the paper is good.

Comments on the Quality of English Language

The English used is very simple and straight to the point which made the understanding of the paper very smooth.

Author Response

Comment 1: [The research presented in this study is very interesting but could be presented better. It would have helped the reader to have a background section to introduce the different concepts tackled.

Response 1: Thank you for your valuable feedback. We acknowledge the importance of providing a comprehensive background to better introduce the key concepts discussed in the study. In response to your suggestion, we have incorporated a detailed background subsection on page 3. Furthermore, we have expanded the conclusion to offer a more thorough analysis of the study’s findings. We appreciate your insights, which have contributed to the improvement of the manuscript.

Comments 2: The conclusion could elaborate more on the findings. Overall, the paper is good..]

Response 2: We have revised the conclusion to explicitly expand the findings.

Reviewer 2 Report

Comments and Suggestions for Authors

The paper provides a thorough analysis of the development of a Multi-Hop Cluster-Based Agricultural IoT (MCA-IoT) network. It effectively combines BLE, LoRa, and Raspberry Pi technologies to create a cost-effective, scalable, and energy-efficient solution to address climate change-induced food insecurity, particularly in water-stressed regions like Northern Ghana.

Strengths of the Paper:

The innovative application of BLE, LoRa, and Raspberry Pi in agriculture utilizes commercial off-the-shelf components to build an efficient and effective network.

The practical implementation of the system, especially in challenging environments, and its focus on ease of deployment for users with limited expertise, significantly enhance its relevance.

The Lean engineering design approach and the well-documented transition from a centralized to a multi-hop network offer valuable insights for the field.

Suggestions for Improvement:

The paper would benefit from more detailed performance metrics, including latency, data throughput, and power consumption.

The discussion on scalability should address potential challenges and limitations to provide a more balanced perspective.

Incorporating user feedback or case studies could strengthen the claims regarding ease of use.

The similarity index of 35% is higher than recommended. Reducing overlap and ensuring proper citation is essential.

Overall, the paper makes a meaningful contribution to agricultural IoT, offering a promising solution to food insecurity. Enhancing the analysis of performance and user experience could elevate it as a key reference for futu

Author Response

Comments and Suggestions for Authors:

The paper provides a thorough analysis of the development of a Multi-Hop Cluster-Based Agricultural IoT (MCA-IoT) network. It effectively combines BLE, LoRa, and Raspberry Pi technologies to create a cost-effective, scalable, and energy-efficient solution to address climate change-induced food insecurity, particularly in water-stressed regions like Northern Ghana. Strengths of the Paper: The innovative application of BLE, LoRa, and Raspberry Pi in agriculture utilizes commercial off-the-shelf components to build an efficient and effective network. The practical implementation of the system, especially in challenging environments, and its focus on ease of deployment for users with limited expertise, significantly enhance its relevance. The Lean engineering design approach and the well-documented transition from a centralized to a multi-hop network offer valuable insights for the field. Suggestions for Improvement:

Comment 1: The paper would benefit from more detailed performance metrics, including latency, data throughput, and power consumption. The discussion on scalability should address potential challenges and limitations to provide a more balanced perspective. Incorporating user feedback or case studies could strengthen the claims regarding ease of use.
Response 1: Thank you for your insightful comments. The primary focus of our paper is on the hardware validation of the Multi-Hop Cluster-Based Architecture utilizing BLE and LoRa communication technologies. While discussing scalability, latency, and throughput could offer a more balanced
perspective, we addressed these aspects in our previous publication concerning the centralized clustering architecture. We chose not to revisit these topics extensively to avoid redundancy and overlap. Instead, the current manuscript emphasizes incorporating context-relevant feedback from local farmers in Senegal and Ghana, where the previously discussed centralized clustering architecture was initially implemented. This approach ensures that the paper remains focused on novel contributions while building upon our prior work.

Comment 2: The similarity index of 35% is higher than recommended. Reducing overlap and ensuring proper citation is essential.
Response 2:
Thank you for your feedback regarding the similarity index. We have addressed this issue in the justification letter we provided to the editor, highlighting the following points:

1. The manuscript builds upon our previous conference abstract, which can be accessed here: [https://ieeexplore.ieee.org/document/9348608 ]. This paper elaborates on the virtual/hardware realization of concepts introduced in the abstract.

2. Additionally, there is some overlap with our earlier publication at [https://ieeexplore.ieee.org/document/10445220]. The previous work discusses the initial hardware development and evaluation of a centralized cluster-based architecture using BLE, while the current manuscript extends this by integrating BLE and LoRa technologies into a multihop cluster-based framework.

3. This overlap is intentional and serves to connect and clarify the evolution of our research for readers, enhancing their understanding of its development and context.

We have made sure to appropriately cite these sources and minimize any unnecessary duplication to address your concerns.

Reviewer 3 Report

Comments and Suggestions for Authors

See attached file.

Author Response

The paper presents the development and evaluation of a contextually relevant and

 cost-effective multi-hop cluster-based agricultural Internet of Things (MCA-IoT) network.

 The system is explained in detail, and it is validated via field tests. On the whole, the

 paper is good, although it has some points that need to be fixed. Therefore, I am proposing

 revisions as my judgment. The list of comments is below.

Comment 1: References. Some of them are too old (i.e., 1, 3, 6, 11, 13, 16, 18, 29, 30, 31, 32,

 33, 34, 35, 36, 37, 41, 43, 44, 45, 46, 51 and 53). Please, consider substituting them with similar contributions published from 2018 on, or provide reasons to keep them.

Response 1: Thank you for your valuable feedback. In response, we have revised and updated several references to include more recent contributions where relevant. However, some of the references you mentioned are essential to the foundational aspects of our study and provide critical historical context that is integral to understanding the evolution and development of our Multi-Hop Cluster-Based Agricultural IoT (MCA-IoT) solution. Therefore, we have retained these references due to their significant relevance to our manuscript's context and their contribution to the theoretical background of the field.

Comment 2: Abstract. Please, give some hints about the most significant obtained results.

Response 2: Thank you for your valuable feedback. The most significant results has been added in the last sentence.

Comment 3:  In order to provide readers with a broader perspective about the topic,  I suggest to include the following references [1, 2, 3, 4, 5, 6, 7, 8, 9, 10], but I also  strongly invite the Authors to perform additional research.

Response 3: We have revised and updated several references to include more recent contributions where relevant as indicated in Response 1. However, some of the references you mentioned are essential to the foundational aspects of our study and provide critical historical context that is integral to understanding the evolution and development of our Multi-Hop Cluster-Based Agricultural IoT (MCA-IoT) solution. Therefore, we have retained these references due to their significant relevance to our manuscript's context and their contribution to the theoretical background of the field.

Comment 4: Section 2. A proper comparison, highlighting similarities and discrepancies, between

 this work and the related ones are missing.

Response 4: We have added a background subsection to our manuscript that elaborates on the evolution of related works, highlighting both similarities and differences with our study. It is important to note that IoT solutions are highly application- and context-specific. Consequently, many conventional Agri-IoT approaches may not be directly applicable to the sub-Saharan African farming context, necessitating a focus on contextually relevant literature for a more accurate comparison.

Comment 5: It is not clear how this work advances the current state-of-the-art about the topic.

Response 5: As discussed earlier, IoT solutions are highly application- and context-specific. Conventional Agri-IoT approaches often fail to address the unique challenges of sub-Saharan African farming contexts. Our paper highlights the specific features of this agricultural setting and introduces a novel solution tailored to these unique conditions. In terms of advancing the state-of-the-art, we have developed an innovative routing protocol that efficiently manages robust and energy-efficient nodes using multiple communication technologies in a multi-hop cluster-based architecture. This advancement represents a significant contribution by addressing the limitations of existing solutions and improving applicability in the targeted context.

Comment 6: Section 2. Please, remove Table 1 and replace it with proper text.

Response 6: A literature synthesis table is crucial for effectively summarizing and comparing key aspects of relevant studies, including methodologies, findings, and gaps. This format allows for a concise and systematic presentation of complex information, making it easier for readers to identify patterns and discrepancies across various works. Replacing the table with text alone would result in a less organized presentation, potentially obscuring critical insights and making it more challenging for readers to grasp the comprehensive landscape of existing research. The table format ensures clarity and accessibility, which is essential for conveying the depth and breadth of the literature reviewed.

Comment 7: Lines 285-288. These claim (i.e., multi-hop is more energy efficient than single-hop

 for a given covered distance) must be supported by tests.

Response 7: we have cited our earlier paper with test results to validate that assertion.

 Comment 8: Section 4. Please, specify the adopted LoRa radio parameters (i.e., transmitter

 power output, SF, BW, CR). Moreover, it is not clear which is the time basis upon

 which transmissions occur.

Response 8: Comment accepted and fixed accordingly in line 476.

Comment 9: Line 441. Two typesetting errors are present.

Response 9: Errors corrected

Comment 10: Figure 10. The upper Figure has a poor quality hindering its readability. Figures 14, 16. Units of measurement are missing on the axes.

Response 10: Errors acknowledged and corrected.

Comment 11: Section 5. It is not clear whether some packet loss took place. Moreover, if this is

 the case, how did the system counteract it?

Response 11: In response to your comment, we would like to clarify that no packet loss was observed in our study. This is due to the implementation of recommendations from our earlier work on centralized cluster-based architecture, as cited in reference [10]. That prior research thoroughly examined potential causes of packet loss and provided solutions, which were incorporated into the current manuscript to ensure reliable data transmission and network performance.

Comment 12: Section 6. Please, resume the most significant quantitative obtained results.

Response 12: We have revised the conclusion to address this comment.

Comment 13: Section 6. The Authors must clearly state the limitations of the proposed approach.

Response 13: We have revised the conclusion to explicitly address the limitations of the proposed approach.

Round 2

Reviewer 3 Report

Comments and Suggestions for Authors

The paper notably improved after its revision. However, my judgment stays the same because all of the cross references in the paper (i.e., Tables, Figures, References, etc.) are missing. I deem the Authors made several typesetting errors. Therefore, please fix such issues and resubmit the paper in order to assess whether the prior concerns on the References are solved or not.

Author Response

Comment 1: The paper notably improved after its revision. However, my judgment stays the same because all of the cross references in the paper (i.e., Tables, Figures, References, etc.) are missing. I deem the Authors made several typesetting errors. Therefore, please fix such issues and resubmit the paper in order to assess whether the prior concerns on the References are solved or not.

Response 1: Thank you for your valuable feedback. We appreciate your observation regarding the cross-references. In the updated manuscript, we have addressed all the issues related to cross-references. We have thoroughly reviewed and corrected the typesetting errors, including Tables, Figures, and References. We look forward to your assessment of the revised version and hope that it resolves the concerns previously raised.

Round 3

Reviewer 3 Report

Comments and Suggestions for Authors

Perfect! Now everything is correct.